# Orthohantavirus infections in humans and rodents in the Yichun region, China, from 2016 to 2021

**Shi-Wen Liu**[1,2], **Jian-Xiong Li**[2], **Long Zou**[1], **Xiao-Qing Liu**[2], **Gang Xu**[2], **Ying Xiong**[2]*, **Zhong-Er Long**[1]*

1 College of Life Sciences, Nanchang Key Laboratory of Microbial Resources Exploitation & Utilization from Poyang Lake Wetland, Jiangxi Normal University, Nanchang, Jiangxi, China, 2 Laboratory of Viral Infectious Disease, Jiangxi Provincial Center for Disease Control and Prevention, Nanchang, Jiangxi, China

* xiongying8300087@163.com (YX); longzhonger@jxnu.edu.cn (Z-EL)

**Data Availability Statement:** All relevant data are within the manuscript and its Supporting Information files.

## Abstract

### Background

Rodents are the predominant natural hosts of orthohantavirus and the source of human infection, hemorrhagic fever with renal syndrome (HFRS) caused by orthohantavirus is a severe public health problem in the Yichun region, Jiangxi Province, China. However, little information is known about the infection of orthohantavirus in humans and rodents, and the genetic characteristics of the epidemic orthohantavirus in the region.

### Methods

The clinical data of HFRS cases in 2016–2021 was analyzed. Virus infection in rodents was analyzed by orthohantavirus antigen detection using immunofluorescent assay, and the species of orthohantaviruses in rodents and patients were identified by real-time RT-PCR and gene sequencing. The S and M segments of orthohantaviruses from rodents and patients were recovered and analyzed.

### Results

A total of 1,573 HFRS cases were reported in the Yichun region from 2016 to 2021, including 11 death cases. HFRS cases peaked twice each year: in winter from November to January and early summer from May to June. Farmers constituted the predominant population suffering from HFRS. The orthohantavirus antigen was identified in five species of rodents: *Apodemus agrarius* (*A. agrarius*), *Rattus norvegicus* (*R. norvegicus*), *Sorex araneus*, *Rattus losea* (*R. losea*), and *Niviventer confucianus* (*N. confucianus*). The real-time RT-PCR test and genetic analysis results showed that Hantaan orthohantavirus (HTNV), Seoul orthohantavirus (SEOV), and Dabieshan orthohantavirus (DBSV) were circulated in the rodents. HTNV, SEOV, and DBSV from the rodents were distantly related to other known orthohantaviruses and belonged to novel genetic lineages. SEOV and HTNV were found in HFRS patients, but 97.8% (90/92) of the infections were caused by HTNV. Winter and early

**Funding:** This work was supported partially by the Key Research and Development Program of Jiangxi Province (No. 20203BBG73055 to YX, URL: http://kjt.jiangxi.gov.cn/), Science and Technology Program of Health Commission of Jiangxi Province (No. 202211313 to SWL, URL: http://hc.jiangxi.gov.cn/), and the Development Grant of Key Laboratory of Health Commission of Jiangxi Province (to YX, URL: http://hc.jiangxi.gov.cn/). The funders had no role in the study design, data collection, analysis, decision to publish, or the preparation of the manuscript.

**Competing interests:** The authors have no conflicts of interest to declare.

summer peaks were both caused by HTNV. The HTNV sequences recovered from HFRS cases were closely related to those from *A. agrarius*.

## Conclusions

In the Yichun region, the orthohantaviruses transmitted in rodents include HTNV, SEOV, and DBSV, which have obvious genetic characteristics and high genetic diversity. At the same time, this region is an HFRS mixed epidemic area dominated by HTNV, with two peaks every year, which deserves our high attention.

### Author summary

HFRS caused by orthohantavirus is a serious public health problem in the Yichun region in Jiangxi Province, China. However, little information is known about the infection of orthohantavirus in humans and rodents, and the genetic characteristics of the epidemic orthohantavirus in the region. To reveal the above problems, we analyzed the epidemiology of HFRS and performed a molecular investigation of orthohantavirus in the Yichun region. During 2016–2021, a total of 1,573 HFRS cases were recorded with an average fatality rate of 0.70%. HFRS consistently occurred and peaked in winter and early summer in the Yichun region, it was a mixed epidemic area of HFRS dominated by HTNV. We identified new genetic variants of SEOV, HTNV, and DBSV from rodent lung samples in the Yichun region, which shows the high genetic diversity of the orthohantaviruses. Special attention should be paid to the high genetic diversity of orthohantavirus when developing vaccines and antiviral drugs. We also found the new genetic variant of HTNV from rodents caused human infection and was the main prevalent virus of human infection in the Yichun region. These results will provide very important information for HFRS control in the Yichun region and other areas in the world.

## Background

Members of the family Hantaviridae, genus Orthohantavirus, orthohantaviruses are enveloped, single-stranded, and negative sense RNA viruses [1]. The orthohantavirus RNA genome consists of three segments: L (large), M (medium), and S (small), which encode RNA-dependent RNA polymerase (RdRp), glycoproteins (Gn and Gc), and nucleocapsid protein (N), respectively [2]. Orthohantaviruses are carried and transmitted by small animals, including rodents, shrews, and bats [3]. Each orthohantavirus species usually appears to be primarily associated with one host species. For instance, HTNV has been demonstrated in *A. agrarius* in Far East Russia, China, and South Korea. SEOV harbored by *R. norvegicus* is present worldwide. Huangpi virus has been found in bats (*Pipistrellus abramus*), and Cao Bang virus in shrews (*Anourosorex squamipes*) in China [4]. Transmission among nature hosts and from hosts to human generally occurs via aerosolized infected excreta. Rodents are the predominant natural hosts and the source of human infection [5]. In rodents, these viruses typically cause asymptomatic persistent infections that cause no apparent harm. However, in humans, rodent-associated viruses cause hemorrhagic fever with renal syndrome (HFRS) in Eurasia and cardiopulmonary syndrome in North and South America [6,7].

 HFRS is characterized by fever, acute kidney damage, and hemorrhage manifestations; the case fatality rates are up to 15% [3,8]. The majority of the HFRS cases in the world occurred in

China. A total of 1,557,622 HFRS cases were recorded in China from 1950 to 2007, the cases peaked in 1986 when the largest annual number (115,804 cases) of HFRS cases was reported [9]. China has comprehensively controlled and prevented HFRS, successfully decreasing the incidence rate from 1990 to 2010 [9,10]. However, it remains a considerable public health threat in China, having 100,000 cases and causing 1,116 deaths over the past 10 years [11]. Moreover, the incidence rate has increased drastically in recent years in some parts of China [12]. Several orthohantavirus species can cause HFRS, including HTNV, SEOV, Puumala orthohantavirus, and Dobrava-Belgrade orthohantavirus [13]. The predominant causative pathogens of HFRS in China were HTNV and SEOV [14].

A total of 7,203 HFRS cases were reported in Jiangxi from 2005 to 2018, and the case fatality rate was 1.34% [15]. More than 50% of these cases occurred in the Yichun region [16]. The Yichun region has implemented the HFRS-targeted Expanded Program on Immunization (EPI) among people aged 16–60 years since 2009. However, little was known about the epidemiology of HFRS after the HFRS-targeted EPI. Despite this situation, no molecular surveys of orthohantaviruses have been carried out in the Yichun region, and the circulating pathogens of HFRS in humans and rodents remained unknown.

The objective of this study was to investigate the epidemiology features of HFRS across the Yichun region from 2016 to 2021 and to clarify the causative pathogens in both humans and rodents. The findings of the study will provide crucial information for developing effective HFRS control strategies, not only in the Yichun region but also in other areas of the world.

## Materials and methods

### Ethics statement

This study was approved by the ethics committee of Jiangxi Provincial Center for Disease Control and Prevention to ensure the patient's informed consent, confidentiality, and anonymity, and written consent was obtained from the patients. Rodents were strictly treated according to the experimental animal welfare and ethical treatment guidelines issued by the Ministry of Science and Technology of China. (SYXK (Gan) 2017–0006).

### HFRS data

Data on reported cases of HFRS from 2016 to 2021 were obtained from the Yichun region, located at 27° 33′–29° 06′ N, 113° 54′–116° 27′ E, in the northwest of Jiangxi Province, China. The information on HFRS cases including age, sex, occupation, and incidence rates was obtained from the China Information System for Disease Control and Prevention (CISDCP). HFRS cases from the CISDCP were reported according to the Diagnostic Criteria for Epidemic Hemorrhagic Fever (WS278-2008) issued by the National Health Commission of the People's Republic of China (https://hbba.sacinfo.org.cn/stdDetail/96a89f9c0869dffc94e84e485a45930c).

### Samples collection

The prevalence, species, and genetic characteristics of orthohantaviruses in rodents were explored. The rodents were trapped in the spring and fall/winter seasons by the night trapping method in HFRS endemic areas in the Yichun region from 2016 to 2021. Rodents were trapped in cages at 5- meter intervals and baited with peanuts. The trap sites included field and residential areas from three counties: Gao'an (28.4178˚N, 115.3753˚E), Tongu (28.52311˚N, 114.37036˚E), and Shanggao (28.23423˚N, 114.92459˚E). The rodents were initially classified into specific species by morphological examination and dissected as soon as they were

authenticated, and lung tissues were collected. The species of rodents that showed the presence of orthohantavirus were confirmed by *cytb* gene analysis according to a previously described method [17].

The species and genetic characteristics of orthohantaviruses in humans were also explored. Serum samples were collected from HFRS patients in the epidemic area in the Yichun region, including Gao'an, Yifeng (28.38555°N, 114.7803°E), Shanggao, and Fengxin (28.6879°N, 115.40036°E) counties. Patients with specific IgM antibodies against orthohantaviruses and within 7 days of onset were selected.

## Detection of orthohantaviral antigen in rodents

The viral infection rate in rodents was analyzed by orthohantaviral antigen detection. Orthohantaviral antigens in rodent lung samples were detected by the immunofluorescent assay as described previously [18]. Lung samples were sliced and mounted on slides by a cryomicrotome, dried in a bio-safety cabinet, fixed with cold acetone for 20 min, and air-dried. FITC-labeled anti-SEOV/HTNV antibodies (10 μL; provided by the Fourth Military Medical University) were added to each sample. The slides were incubated in a 37°C water bath for 45 min, washed with 1× PBS buffer three times, and then distilled water once, blow-dried, and sealed. The slides were observed under a fluorescence microscope at 100× magnification.

## Extraction of total RNA

The rodent lung samples with the orthohantaviral antigen that was confirmed by immunofluorescent assay were mechanically homogenized by a bullet blender (Next advance in the United States) and then centrifuged at 8000 *g* for 10 min. The supernatant of each tissue suspension was collected and used for nucleic acid extraction. Total RNA from supernatants and patient serum was extracted with QIAamp viral RNA mini kit (QIAGEN). Nucleic acid was dissolved with 40 μL of nuclease-free water and stored at −80°C.

## Real-time RT-PCR

Total RNA from supernatants and patient serum was initially detected by real-time RT-PCR for virus species identification. Real-time RT-PCR for HTNV and SEOV was performed as previously described [19]. The primers and probes of real-time RT-PCR were HTNV-forward, HTNV-reverse, HTNV-Probe, SEOV-forward, SEOV-reverse, and SEOV-Probe (S1 Table). Total RNA from rodents and patients was reversed and quantitatively analyzed using an AgPath-ID one-step RT-PCR kit (Thermo Fisher Scientific, USA).

All primers and probes were provided by Sangon Biotech Co, Ltd (Shanghai, China). Real-time RT-PCR reactions were performed on an ABI7500 system (Applied Biosystems Instruments, USA). The procedures were as follows: reverse transcription at 45°C for 10 min, initial denaturation at 95°C for 10 min, 45 cycles at 95°C for 15 s, and 55°C for 45 s. The sample with a Ct value of less than 40 for at least 1 of the replicates was considered positive for orthohantavirus.

## The M and S segments sequencing

The M and S segments of orthohantavirus were amplified from rodents with the orthohantaviral antigen and patient serum with orthohantavirus. PCR products were sequenced for virus species identification and genetic characteristics analysis.

Partial M segments were amplified by a nested PCR protocol including two amplification rounds. A PrimeScript one-step RT-PCR kit (TAKARA, Dalian) was used in the first round,

**Table 1. The cases number and morbidity rate of HFRS in the Yichun region (2016–2021).**

| Year | No. of cases | No. of deaths | Morbidity (per 100,000) | Mortality (per 100,000) | Fatality (%) |
|------|--------------|---------------|-------------------------|-------------------------|--------------|
| 2016 | 311 | 1 | 5.64 | 0.018 | 0.32 |
| 2017 | 278 | 1 | 5.02 | 0.018 | 0.36 |
| 2018 | 352 | 5 | 6.34 | 0.09 | 1.42 |
| 2019 | 288 | 1 | 5.17 | 0.018 | 0.35 |
| 2020 | 218 | 3 | 3.91 | 0.054 | 1.38 |
| 2021 | 126 | 0 | 2.25 | 0 | 0 |

and procedures were as follows: an initial step for cDNA synthesis at 50˚C for 30 min, denaturation at 94˚C for 2 min, 35 cycles of 94˚C for 30 s, 52˚C for 30 s, 72˚C for 1 min, and final extension at 72˚C for 10 min. Total RNA was used as the template with primers HV-F and HV-R (S1 Table). Amplification was performed in 25 μL volumes each containing 12.5 μL of buffer, 1 μL of enzyme mix, 5 μL of template, 0.5 μL for each primer (20 μmol/L), and 5.5 μL of nuclease-free water. The products of the first round were used as templates for the second round of amplification using 2× GoTaq Green Master Mix (Promega, USA) with the primers, HTNV-F, HTNV-R, SEOV-F, and SEOV-R (S1 Table) [20]. PCR reactions were performed in 50 μL volumes, containing 25 μL of GoTaq Green Master Mix, 5 μL of template, 1 μL of each primer (20 μmol/L), and 18 μL of nuclease-free water. The procedures were as follows: denaturation at 95˚C for 2 min; 35 cycles of 95˚C for 30 s, 55˚C for 30 s, and 72˚C for 40 s; and final extension at 72˚C for 5 min.

The complete S segment was amplified using PrimeScript one-step RT-PCR kit (Takara, Dalian, China) with the primers HV-SF and HV-SR (S1 Table). PCR reactions were performed in 50 μL volumes each containing 25 μL of buffer, 2 μL of enzyme mix, 5 μL of template, 1 μL of each primer (20 μmol/L), and 16 μL of nuclease-free water. RT- PCR was performed at 50˚C for 30 min for cDNA synthesis, denaturation at 94˚C for 2 min; 35 cycles of 94˚C for 30 s, 55˚C for 30 s, and 72˚C for 1 min and 40 s; and final extension at 72˚C for 10 min. The PCR products of the complete S segment were purified and ligated into pESI-T vectors and transformed into *Escherichia coli* DH5α competent cells according to the manufacturer's instructions (TaKaRa, Dalian, China). The presence of an insert was confirmed by PCR and agarose gel electrophoresis. The recombinant pESI-T vector was extracted by SK8191 UNIQ-10 plasmid extraction kit (Sangon, Shanghai, China) according to the manufacturer's instructions.

Partial S segment from samples that failed in complete S segment sequencing was amplified by a nested PCR using the primers HV-pSF, HV-pSR for the first round and HTNV-pSF, HTNV-pSR, SEOV-pSF, SEOV-pSR (S1 Table) for the second round. The PCR protocols were the same as partial M segment amplification.

The PCR products of the partial M, S segment, and recombinant pESI-T vector were sequenced by Sangon Biotech Co. (Shanghai, China).

## Data and genetic analysis

Data of HFRS patients and rodents were analyzed with Excel 2007. Statistical analysis was performed using IBM SPSS Statistics software, version 21. The inter-group difference in the virus infection rates was compared by using the Chi-square test. A value of $P<0.05$ was considered significant.

The genomic sequences were assembled with Seqman in Lasergene software. Sequences were aligned using the ClustalW method in MEGA X. A sequence identity matrix was obtained with BioEdit version 7.2.5. Phylogenetic analysis was performed using the neighbor-joining method in MEGA X with a bootstrap analysis of 1000 iterations [21].

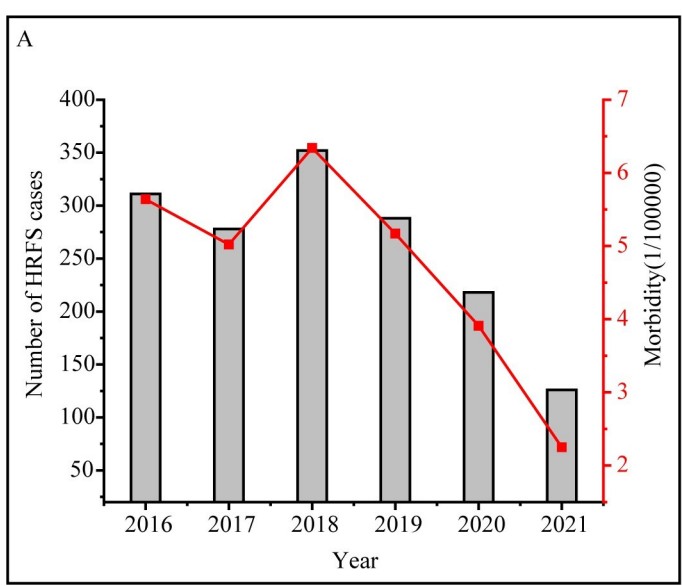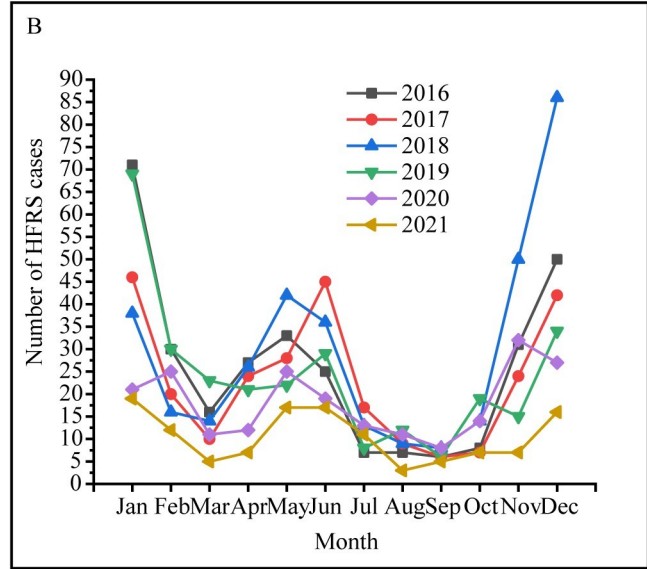

**Fig 1.** (A) The number of cases and the morbidity of HFRS, (B) Monthly distribution of HFRS cases reported, in the Yichun region from 2016 to 2021.

## Results

### Epidemiological characteristics of HFRS

In 2016–2021, a total of 1,573 HFRS cases were reported in the Yichun region, including 11 death cases. The morbidity ranged from 2.25 to 6.34 per 100,0000 persons annually (Table 1), and a declining trend was observed after 2018 (Fig 1A). The mortality ranged from 0.018 to 0.09 per 100,0000 persons annually (Table 1), and the average fatality rate was 0.70%. HFRS peaked twice each year (Fig 1B). The winter peak occurred from November to January, and the early summer peak occurred from May to June. The peak in winter was higher than that in summer. Of the 11 deaths, three occurred in January, two occurred in November, two occurred in December, and one case per month occurred in March, May, June, and September.

### Demographic distribution of HFRS cases

The demographic distribution of HFRS cases in the Yichun region was shown in Table 2, and it can be seen that 63.4% (998/1573) of the cases were males, and thus the male-to-female ratio was 1.74:1. The top age group in terms of case number was 46–60 years old (Tables 2 and S2), then followed by the over 60 years old group. Regarding occupations, farmers accounted for 65.3% (1027/1573), students accounted for 12.5%, and the unemployed population accounted for 8.5% (Table 2).

**Table 2. Demographic distribution of HFRS cases in the Yichun region (2016–2021).**

| Groups | Sex | | Age (years) | | | | | Occupation | | | |
|---|---|---|---|---|---|---|---|---|---|---|---|
| | Male | Female | <16 | 16–30 | 31–45 | 46–60 | >60 | Farmers | Students | Unemployed population | Others |
| **Number of cases** | 998 | 575 | 223 | 172 | 312 | 455 | 411 | 1027 | 197 | 134 | 215 |
| **Proportion in groups (%)** | 63.4 | 36.6 | 14.2 | 10.9 | 19.9 | 28.9 | 26.1 | 65.3 | 12.5 | 8.5 | 13.7 |

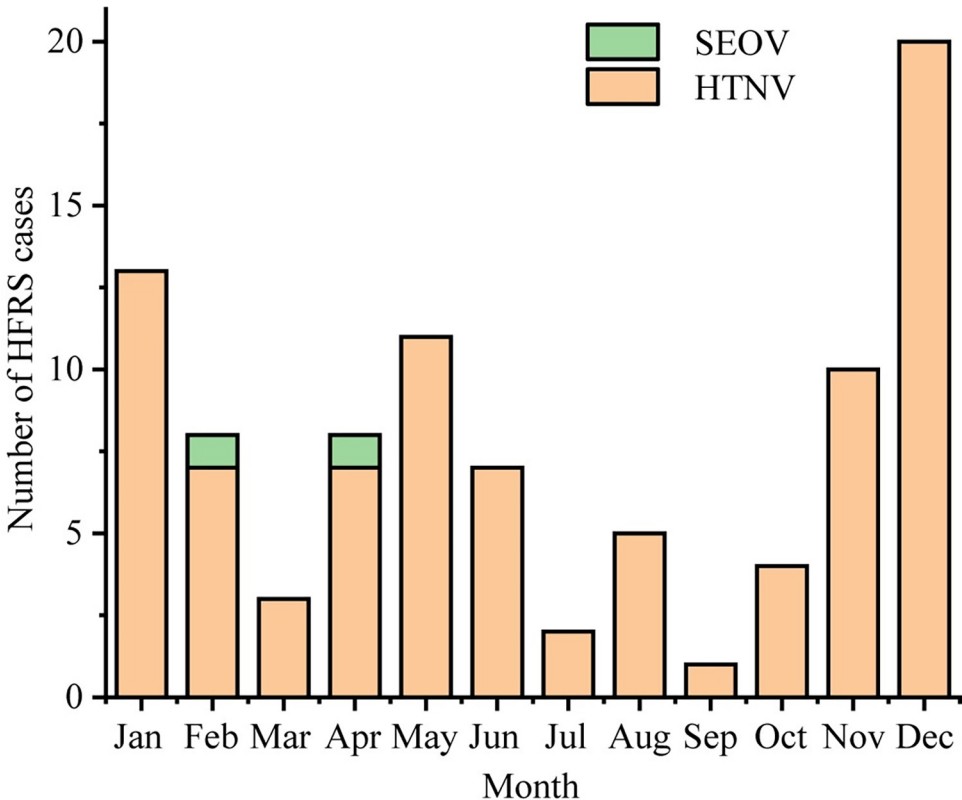

**Fig 2. Monthly distribution of orthohantaviruses from 92 patients.**

### Species of orthohantaviruses in patients

Ninety-two serum samples from 92 patients (S3 Table) were collected from 2017 to 2020. The results of real-time RT-PCR showed that approximately 97.8% (90/92) of HFRS cases were caused by HTNV; and 2.2% (2/90), by SEOV. The onset times of the patients were distributed over 12 months each year. HFRS cases infected with HTNV occurred throughout the year, and the two cases of SEOV occurred in February and April (Fig 2). Two peaks were observed, in May and December, respectively, which were caused by HTNV (Fig 2).

### Virus infection in rodents

From 2016 to 2021, a total of 1,744 rodents were trapped in the Yichun region (Table 3), and the predominant rodent species were *A. agrarius* (40.2%, 701/1744) and *R. norvegicus* (34.9%, 609/1744). The results of the immunofluorescent assay showed that 33 of 1,744 trapped rodents were positive for orthohantaviral antigen, accounting for 1.89%, including 15 *A. agrarius*, 9 *R. norvegicus*, 2 *R. losea*, 3 *Sorex araneus*, and 4 *N. confucianus*, the others were negative to orthohantaviral antigen. The virus infection rate in rodents showed a rising trend after 2018 (S1 Fig) and reached 3.5% (8/228) in 2021. Among 33 rodents with orthohantaviral antigen in lung samples, a total of 20 rodents were confirmed with the *cytb* gene sequencing methods for species identification, the other 13 rodents were not analyzed for lack of samples. The results were consistent with that of the morphological identification, and their *cytb* gene sequences were deposited into GenBank under accession no. OQ187773-OQ187792 (S5 Table).

No significant difference (*P* = 0.473) in virus infection rate was found between *A. agrarius* and *R. norvegicus*. The virus infection rates were 1.37% (10/728) in residential rodents and

**Table 3. Orthohantavirus infection in trapped rodents in the Yichun region (2016–2021) *.**

| Species | Year | | | | | | Total |
|---|---|---|---|---|---|---|---|
| | 2016 | 2017 | 2018 | 2019 | 2020 | 2021 | |
| *A. agrarius* | 84/1 | 107/4 | 130/0 | 165/1 | 148/4 | 67/5 | 701/15 |
| *R.norvegicus* | 90/1 | 114/1 | 149/2 | 132/1 | 44/2 | 80/2 | 609/9 |
| *R. losea* | 17/0 | 33/1 | 40/0 | 29/0 | 17/0 | 59/1 | 195/2 |
| *Mus musculus* | 10/0 | 5/0 | 15/0 | 29/0 | 7/0 | 52/0 | 118/0 |
| *Sorex araneus* | 0/0 | 7/1 | 21/0 | 14/2 | 8/0 | 15/0 | 65/3 |
| *Rattus flavipectus* | 0/0 | 0/0 | 0/0 | 4/0 | 19/0 | 5/0 | 28/0 |
| *N. confucianus* | 0/0 | 2/2 | 8/0 | 3/0 | 5/2 | 10/0 | 28/4 |
| Total | 201/2 | 268/9 | 363/2 | 376/4 | 248/8 | 228/8 | 1744/33 |

*Data showing the number of rodents trapped/number of antigen-positive. The total traps from 2016 to 2021 were 9600, 8000, 8600, 6810, 6740, and 8270, respectively.

2.26% (23/1016) in field rodents (Table 4), showing no significant difference ($\chi2 = 1.810$, $P > 0.05$). The virus infection rate in rodents in spring and fall/winter were 1.99% (18/905) and 1.79% (15/839), respectively (Table 4), showing no significant difference ($\chi2 = 0.95$, $P > 0.05$). However, the infection rates for *R. norvegicus* in spring (8/310, 2.58%) and fall/winter (1/299, 0.33%) differed significantly ($\chi2 = 5.274$, $P = 0.038$).

In 33 orthohantavirus antigen-positive samples, the virus species were identified in 25 rodent samples (Table 4). HTNV and SEOV were identified by real-time RT-PCR, and DBSV was identified by RT-PCR and sequencing. The complete S segment of DBSV had been amplified using the primers HV-SF and HV-SR, and the partial M segment of DBSV had been amplified using the primers HTNV-F and HTNV-R primers (S1 Table). Eight samples failed in the virus species identification because of a lack of samples or improper storage before the reception in our laboratory. HTNV and DBSV were identified in *A. agrarius and N. confucianus*, respectively. SEOV was identified in *R. norvegicus*, *R. losea*, and *N. confucianus*.

## Genetic characteristics of orthohantaviruses from rodents and humans

In total, 17 complete S segments and 18 partial M segments were obtained from 20 rodents (S5 Table). On the other hand, 35 S segments (including 12 complete S and 23 partial S) and 48 partial M segments from 48 patients infected with HTNV, and 2 partial S segments from 2 patients infected with SEOV were recovered (S5 Table).

**Table 4. Orthohantvirus in rodents by species in the Yichun region (2016–2021) *.**

| Species | Area | | Time | |
|---|---|---|---|---|
| | Field | Residential area | Spring | Fall/Winter |
| *A. agrarius* | 701/15/12 (HTNV) | 0/0/0 | 343/8/7 (HTNV) | 358/7/5 (HTNV) |
| *R. norvegicus* | 49/1/1 (SEOV) | 560/8/7 (SEOV) | 310/8/8 (SEOV) | 299/1/0 |
| *R. losea* | 177/2/1 (SEOV) | 18/0/0 | 107/1/1 (SEOV) | 88/1/0 |
| *Mus musculus* | 118/0/0 | 0/0/0 | 64/0/0 | 54/0/0 |
| *Sorex araneus* | 53/2/0 | 10/1/0 | 37/0/0 | 26/3/0 |
| *Ratts flavipectus* | 26/0/0 | 2/0/0 | 25/0/0 | 3/0/0 |
| *N. confucianus* | 17/3/3 (2 DBSV+1 SEOV) | 11/1/1 (DBSV) | 18/1/1 (SEOV) | 10/3/3 (DBSV) |
| Total | 1016/23/17 | 728/10/8 | 905/18/17 | 839/15/8 |

* Data shows the number of rodents trapped/number of antigen-positive/number of PCR-positive (virus species).

Genetic analysis showed that 19 of the complete S segments of HTNV recovered from 7 *A. agrarius* and 12 patients shared 95.0%–100% nucleotide identities. They shared a higher number of nucleotide identities with HTNV strain AYW89-15 from Jiangxi (94.3%–98.6%) province than with HTNV strains from other areas (82.3%–85.6%). Homology analysis of deduced amino acids sequence of the complete S CDS area (429 aa)revealed that the 19 HTNV strains had 95.8%-99.3% similarity with the other HTNV reference strains, and there were 11 to 16 amino acid variations compared to HTNV standard reference strain 76–118. The nucleotide identities among 42 partial S segments of HTNV from 7 *A. agrarius* and 35 patients was 93.2–100%. Seven complete S segments of SEOV recovered from six *R. norvegicus* and one *R. losea* in the Yichun region shared 95.0%–100% nucleotide identities with one another. They shared 87.2%–90.6% nucleotide identities and 95.3%-99.7% amino acid identities with other known SEOV strains. Due to the low viral load in the patient infected with SEOV, only two S segments (SGHu01/2020, GAHu11/2017) of a 104bp length from two patients were obtained. The nucleotide identities analysis based on the partial S segment showed that SGHu01/2020 and GAHu11/2017 shared 100% nucleotide identities with SEOV strain JiangxiXinjianRn-07-2011 and JUN5-14, respectively. However, in this study, they shared only 85.4%-88.3% nucleotide identities with 7 SEOV strains from rodents in the Yichun region. Three complete S segments of DBSV recovered from *N. confucianus* shared 99.7%–100% nucleotide identities and 83.8%–89.6% with other known DBSV strains. The amino acid similarity among the DBSVs in the Yichun region and the reference strains was 97.9%-99.5%.

Fifty-eight partial M segments of HTNV recovered from 10 *A. agrarius* and 48 patients shared 92.3%–100% nucleotide identities with one another, 92.9%–99.3% with strain AYW89-15, and 79.2%–88.6% with HTNV strains from other areas. Analysis of the partial M segment (amino acid positions from 655 to 753) of HTNV showed that 58 strains in this study shared 89.9%-98.9% amino acid identities with other known HTNV strains and had a unique amino acid variation (Ser700Gly). Five partial M segments of SEOV in the Yichun region showed 92.6%–100% with one another and 83.6%–88% with other known SEOV strains. The amino acid similarity of the partial M segment among the SEOVs in this study and other known SEOV strains was 96.9%-100%. Three partial M segments of DBSV shared 85.6%–89.9% nucleotide identities and 98.9%-100% amino acid identities with other known DBSV strains.

Phylogenetic trees were constructed with the S and M segments of orthohantaviruses (Figs 3, 4 and S2). Identical sequences in this study were excluded. In the partial S segment tree (Fig 3), 29 sequences in the Yinchun region were shown, including 19 of HTNV, 8 of SEOV, and 2 of DBSV. Nineteen sequences of the HTNV strains recovered from humans and rodents belonged to the new lineage of the AYW89-15 strain, a new genetic variant found in another part of Jiangxi province [22]. Only one HTNV strain (FXHu02/2018) belonged to the cluster of the AYW89-15 strain, the other HTNV strains from humans and rodents formed a distinct new cluster (Fig 3A). Six SEOV sequences from rodents in the Yichun region belonged to two new lineages and were distantly related to other SEOVs in the S segment tree (Fig 3B). The gene sequences from two patients infected with SEOV were more related to the widely prevalent SEOV strains than that circulated in rodents in this area (Fig 3B). Two DBSV sequences in this study belonged to a new lineage (Fig 3B). The DBSVs in the Yichun region were distantly related to those from Shaanxi, Zhejiang, and Anhui Provinces, China (Fig 3B).

Fig 4 shows 17 sequences from humans and 10 sequences from rodents in the Yichun region in the M segment tree, including 21 of HTNV, 4 of SEOV, and 2 of DBSV. The phylogenetic tree based on the partial M segment sequences showed a clustering pattern similar to that of the tree based on the partial S segment sequences. Analysis of the HTNV partial M segment revealed that 21 HTNV sequences in this study belonged to the lineage of the AYW89-15 strain. FXHu02/2018 from Fengxin county belonged to the AYW89-15 strain cluster, and the

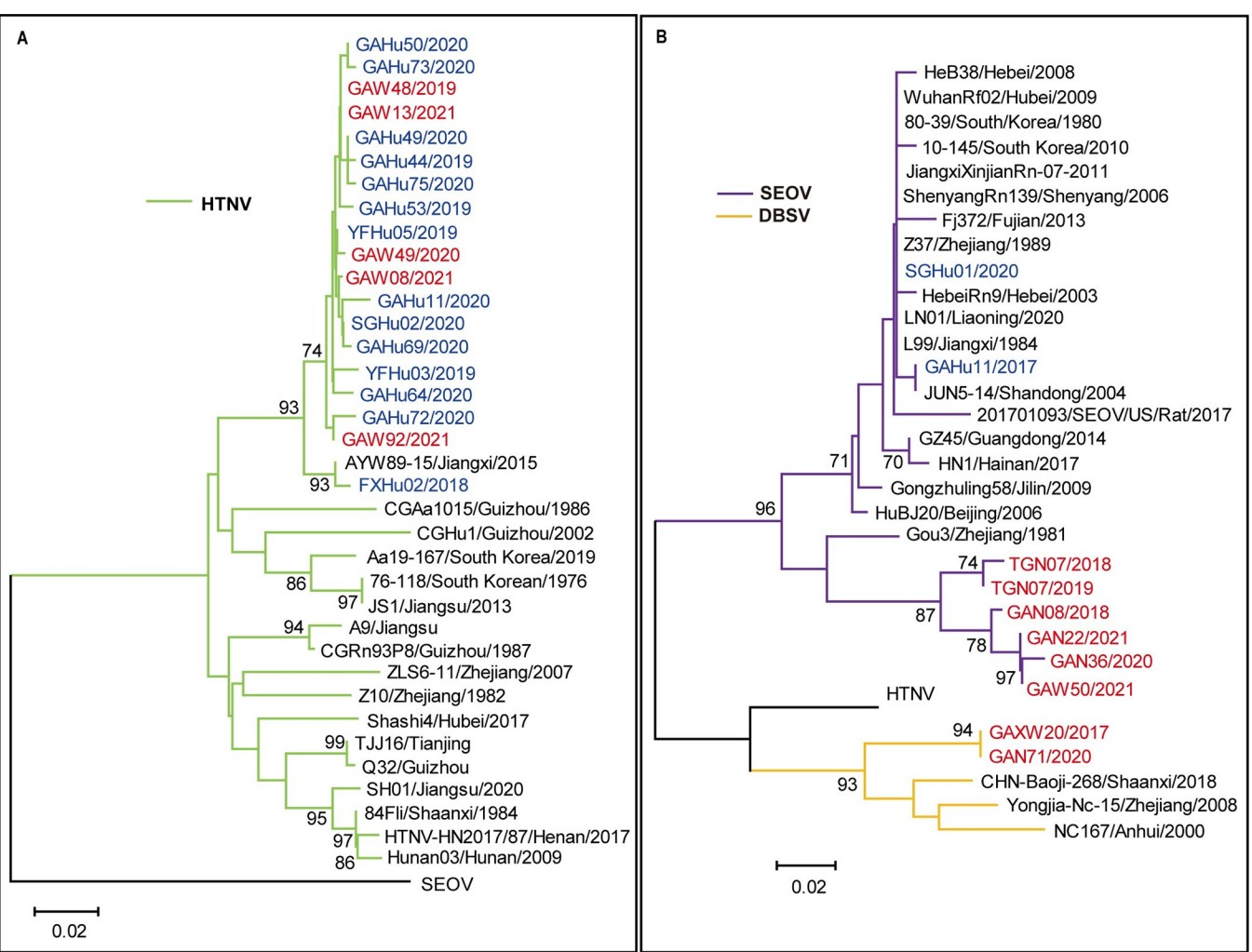

**Fig 3. (A) Phylogenetic tree of orthohantavirus based on partial S segment (634 nt-855 nt), (B) Phylogenetic tree of orthohantavirus based on partial S segment (1070 nt-1173 nt).** The blue font indicates the virus strains recovered from human specimens in the Yichun region; the red font indicates the virus strains recovered from rodent specimens in the Yichun region. The black font indicates the virus strains (S4 Table) from GenBank. Numbers at nodes indicate bootstrap values and only >70% are shown. The scale bars indicate 0.02 substitutions per sit.

other 19 sequences from Gao'an and Yifeng counties formed a novel cluster. In the M tree, the SEOV sequences from rodents are clustered into two separate new lineages (Fig 4). The DBSV sequences in this study in Fig 4 formed a new lineage.

## Discussion

From 2016 to 2021, the morbidity of HFRS in the Yichun region showed a trend of increasing initially and then decreasing, exhibiting a peak in 2018. The lower epidemic of HFRS in 2020 and 2021 might be due to the strict control measures for COVID-19 [23]. Humans can become infected with orthohantaviruses through the inhalation of aerosolized excreta from infected rodents. The most effective way to control orthohantavirus infection is to reduce human exposure to infected rodents and excrement. With the improvement in housing conditions, rodents are no longer commonly found inside people's houses in this area. As a result, outdoor exposure is the most probable way to be infected. The control measures and advice for COVID-19, such as staying home, opening windows, wearing a mask when going out, and cleaning hands

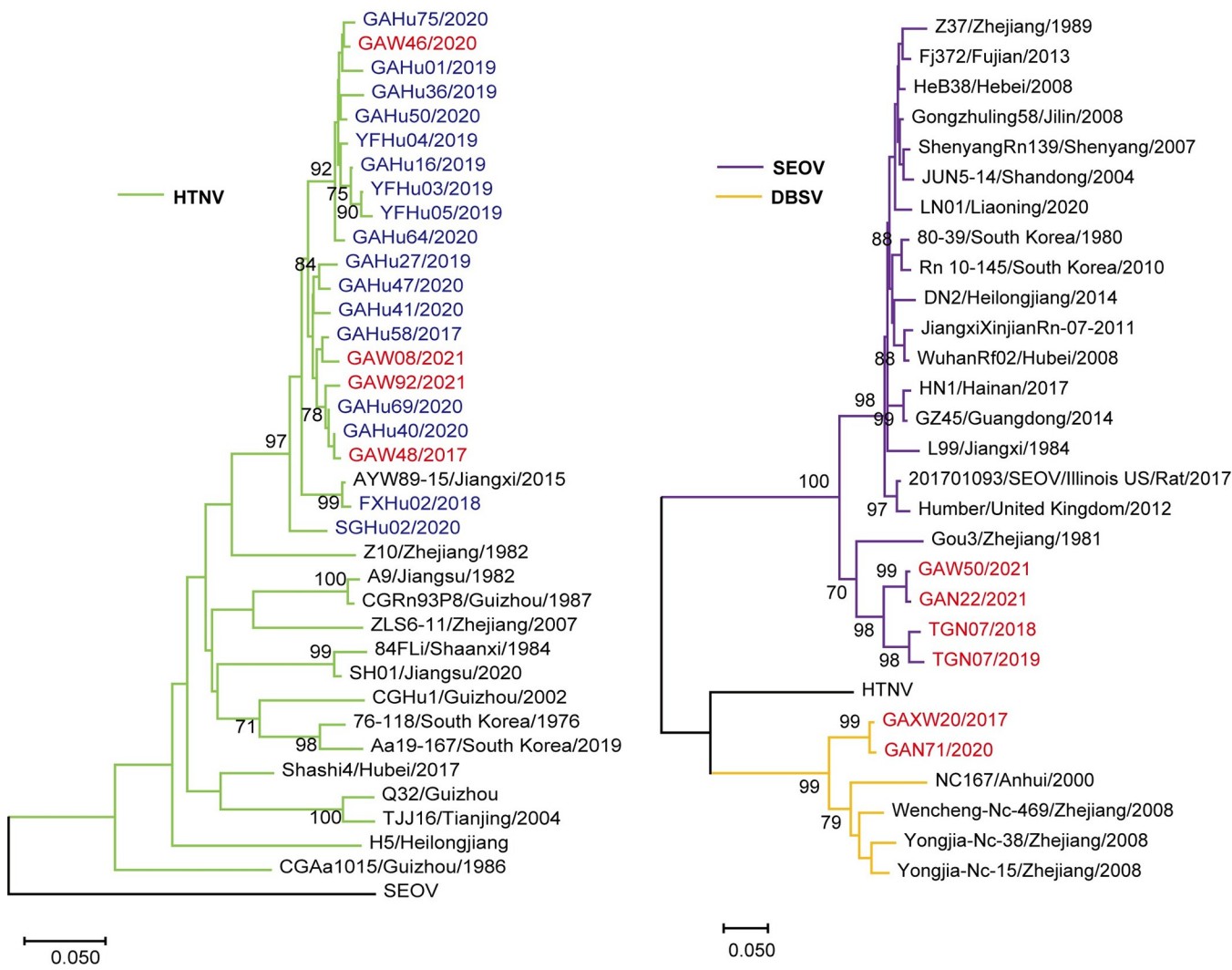

**Fig 4. Phylogenetic tree of orthohantavirus based on partial M segment (2002 nt–2301 nt).** The blue font indicates the virus strain recovered from human specimens in the Yichun region; the red font indicates the virus strain recovered from rodent specimens in the Yichun region. The black font indicates the virus strain (S4 Table) from GenBank. Numbers at nodes indicate bootstrap values and only >70% are shown. The scale bars indicate 0.05 substitutions per site.

frequently could have reduced the inhalation not only of SARS-CoV-2 but also of orthohantaviruses. Given the increasingly optimized control measures for COVID-19 and the restoration of people's normal lifestyle, combined with the rising trend of virus infection rate in rodents after 2018, more HFRS cases may occur in the next few years, so more attention should be paid to the epidemic of HFRS in the Yichun region.

HFRS cases occurred monthly with a major peak in winter and a minor peak in early summer in the Yichun region. Studies showed that the peak of HFRS associated with SEOV occurred in spring, whereas the peak of HFRS associated with HTNV occurred mainly in winter [24]. However, HTNV was the sole source of HFRS in Xi'an and Baoji, China, winter and summer peaks of HFRS cases had been reported [17,25]. In the present study, the winter and early summer peaks were both caused by HTNV. We believed that the characteristics of peaks were not due to the differences in virus species, but rather to the intensity of exposure and climate [26,27]. In this study, farmers were the predominant population and the Yichun region is

a large agricultural region in Jiangxi Province with a subtropical monsoon climate. *A. agrarius*, the major carrier of HTNV, was the predominant rodent in the wild. From spring to late autumn, local farmers spend most of their time planting cereals and vegetables in the field, favoring higher chances of contacting *A. agrarius* and inhaling their body fluids containing HTNV. With an incubation period ranging from 7 to 60 days, the peaks of HFRS can occur in early summer and winter. As for the climate reason, according to the research by Saana Sipari et al [27], climate change can alter the transmission intensity within the host population, thus may influence human infection and changing the peaks. Of course, the climate-induced mechanisms are poorly understood.

Male patients with HFRS were higher in numbers in all age groups. The reason for the difference in incidence rate among different genders may be the gender difference in daily activities. Males have more opportunities to work in farmlands, favoring a higher chance of contacting rodents, while females often stay home to care for their families. The top age groups for the number of cases were 46–60 years old, and then followed by the over 60 years group. This may be due to the tremendous number of young people leaving rural areas to work in cities, while middle-aged and elderly adults tend to stay in villages and engaged in more agricultural activities making them more susceptible to HFRS infection. Hence, effective vaccination and health education should be strengthened among male farmers and protective measures (hats, boots gloves, goggles, face masks, coveralls, et al) in agricultural operations should be taken to reduce the incidence rate of HFRS.

According to the survey results covering 2016–2021, HTNV, SEOV, and DBSV were found cocirculated in rodents in the Yichun region. HTNV and DBSV were found in *A. agrarius* and *N. confucianus*, respectively. SEOV was identified in *R. norvegicus*, *R. losea*, and *N. confucianus*, showing the wide hosts of SEOV. No host spill-over event was found in this study. The infection rates for *A. agrarius* with HTNV and *R. norvegicus* with SEOV showed no significant difference. Although *R. norvegicus* was the predominant rodent in the residential area and the major carrier of SEOV. Only 2.2% (2/92) patients were found infected with SEOV, revealing that the Yichun region is a mixed epidemic area of HFRS dominated by HTNV. One possible reason is that the genetic variants of SEOV from rodents may have weaker pathogenicity. Because the gene sequences from the two patients infected with SEOV were more closely related to the widely prevalent SEOV strains than the new variants of SEOV circulating in rodents in this area. Another possible reason is that the symptom of SEOV infection is mild and is thus often neglected [15,28]. We should strengthen the surveillance of SEOV in patients to determine its epidemic status in humans and research is needed to learn more about the pathogenicity of SEOV genetic variants circulating in rodents in the Yichun region. The two cases of HFRS infected by SEOV in this study occurred in spring (February and April, respectively), thus may relate to the higher infection rates in *R. norvegicus* in the spring.

Phylogenetic analysis was performed for orthohantaviruses from rodents and humans in the study and the phylogenetic trees (Figs 3, 4 and S2) were constructed. At first, the Neighbor-joining method and the Maximum likelihood method were applied to construct the phylogenetic trees, and the resulting phylogenetic trees were similar. Because all the sequences analyzed belong to orthohantavirus with a small evolutionary distance, and the length of the gene sequence analyzed is short, especially for some partial S segments, the neighbor-joining method was ultimately chosen for phylogenetic analysis. It is also worth noting that some strains, such as strain SGHu02/2020, have different locations on S and M segment phylogenetic trees, this could be due to the relative conservatism of the S segment, and the sequence of the S segment we analyzed is shorter (222 nt) than the M segment (298 nt), with fewer nucleotide variations. Another possible reason is that the orthohantavirus RNA genome consists of three independent segments, and the segment recombination events could happen in nature.

The S or M segment of SGHu02/2020 may have undergone a recombination event. However, the gene sequences obtained were limited, and a conclusion on the recombination event can not be drawn yet. In the future, more attention will be paid to the recombination events.

At least nine genetic lineages of HTNV are endemic in East Asia, including China [29]. HTNV strains belonging to novel lineages have been reported in China, including AWY89-15, which was isolated from other places in Jiangxi province [22,30,31]. In the current study, all the HTNV strains in the Yichun region belonged to the lineage of the new genetic variants strain AWY89-15, at the same time most of them formed new clusters (Fig 4). These results suggested the unique characteristic and genetic diversity of HTNV in the Yichun region. The HTNV sequences recovered from HFRS cases were closely related to those from *A. agrarius*, suggesting that the new genetic variant of HTNV is directly transmitted from rodents to humans. This is the first evidence of the new genetic variant of HTNV infecting humans and becoming the main prevalent virus of human infection in the Yichun region. It is necessary to conduct research on the protective efficiency of vaccines due to the prevalence of new genetic variants in rodents and humans in this area.

SEOV is a worldwide endemic orthohantavirus [32]. Most known SEOV strains are genetically homogeneous and do not show geographic clustering, and few SEOV strains differed from most SEOV strains by up to 15% in nucleotides [1]. The SEOV strains from rodents in this study clustered together and formed two new distinct lineages in the phylogenetic trees. The sequences of these genetic variants shared low identity ($\leq$88.0% nucleotide for the complete S segment and $\leq$90.2 nucleotides for the partial M segment) with known SEOV strains, including those (L99, JiangxiXinjianRn-07-2011) from other parts of Jiangxi Province. The SEOV sequences recovered in the Yichun region enlarged the genetic diversity of SEOV.

DBSV, first identified in the Anhui province of China, is a novel species in the genus *Orthohantavirus* [20]. It is carried by *N. confucianus*, which is widely distributed in China. The virus has been detected in *N. confucianus* in Yunnan, Zhejiang, and Shaanxi Provinces, China [26,33,34]. In the present study, DBSV was detected in *N. confucianus* in 2017 and 2020, indicating that DBSV continued to be prevalent in the Yichun region. The identification of DBSV in the Yichun region enlarged its geographical distribution. Given that *N. confucianus* is widely distributed in China, DBSV will probably be found in other places in China. The pathogenicity of DBSV is unclear, attention should also be paid to DBSV for its high similarity with HTNV and wide geographical distribution [25].

A limitation of this study is that we attempted to amplify orthohantaviral genome sequences from all antigen-positive rodents or human serum samples. Some were not successfully amplified probably because of a lack of samples, a long delay of test after the date of onset, low virus load, and improper storage before the reception in our laboratory.

## Conclusion

This work showed that HFRS consistently occurred in the Yichun region and peaked in the summer and winter. Three orthohantaviruses: HTNV, SEOV, and DBSV circulated in the rodents and belonged to novel genetic lineages. The Yichun region was a mixed epidemic area of HFRS dominated by HTNV. The HFRS epidemic in the Yichun region should be closely monitored due to its distinct genetic characteristic and high genetic diversity.

## Supporting information

**S1 Table. Primers and probes used in this study.**
(XLSX)

**S2 Table. The age distribution of HFRS cases reported in the Yichun region from 2016 to 2021.**
(XLSX)

**S3 Table. The information on 92 HFRS cases.**
(XLSX)

**S4 Table. Reference sequences of orthohantavirus used for analysis in this study.**
(XLSX)

**S5 Table. Sequence information on orthohantavirus obtained in this study.**
(XLSX)

**S1 Fig. The virus infection rate in rodents.**
(TIF)

**S2 Fig. Phylogenetic tree of orthohantavirus based on the CDS region (1290 nt) of S segment.** The blue font indicates the virus strain recovered from human specimens in the Yichun region; the red font indicates the virus strain recovered from rodent specimens in the Yichun region. The black font indicates the virus strain from GenBank. Numbers at nodes indicate bootstrap values and only >70% are shown. The scale bars indicate 0.05 substitutions per sit.
(TIF)

## Acknowledgments

We thank all the municipality CDCs of the Yichun region for the investigation of the HFRS epidemic. We also thank Jun Zhou, Yanni Zhang, Yun Xie, and Yong Shi for their help with data collection and experiments.

## Author Contributions

**Formal analysis:** Shi-Wen Liu, Ying Xiong, Zhong-Er Long.

**Funding acquisition:** Shi-Wen Liu, Ying Xiong.

**Investigation:** Shi-Wen Liu, Xiao-Qing Liu, Gang Xu.

**Methodology:** Shi-Wen Liu, Jian-Xiong Li, Long Zou, Xiao-Qing Liu, Ying Xiong.

**Project administration:** Jian-Xiong Li, Ying Xiong, Zhong-Er Long.

**Resources:** Jian-Xiong Li, Ying Xiong.

**Software:** Jian-Xiong Li, Gang Xu.

**Supervision:** Jian-Xiong Li, Ying Xiong, Zhong-Er Long.

**Validation:** Long Zou, Xiao-Qing Liu.

**Visualization:** Shi-Wen Liu, Jian-Xiong Li, Long Zou.

**Writing – original draft:** Shi-Wen Liu, Xiao-Qing Liu, Gang Xu.

**Writing – review & editing:** Shi-Wen Liu, Long Zou, Ying Xiong, Zhong-Er Long.

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
