## [Decision Letter · Decision Letter 0]

1 Mar 2023

Dear Ms Liu,

Thank you very much for submitting your manuscript "Orthohantavirus infections in humans and rodents in the Yichun region, China, from 2016 to 2021" for consideration at PLOS Neglected Tropical Diseases. As with all papers reviewed by the journal, your manuscript was reviewed by members of the editorial board and by several independent reviewers. In light of the reviews (below this email), we would like to invite the resubmission of a significantly-revised version that takes into account the reviewers' comments. 

Both reviewers have requested substantial additional details for methodology, which would be necessary for a proper evaluation of the results after revision. Adding more detail in the discussion of the biology of the hosts and transmission in the introduction and discussion would also help to strengthen the paper. Reviewer 2 has provided more detailed comments directly in the manuscript file, so please ensure that you review those carefully. The authors do not need to strictly adhere to every suggestion about which sentences belong in which sections, but please consider the revisions outlined. Please also consider asking a native english speaker to revise the manuscript for grammar, as suggested by reviewers. 

Reviewer 1's additional comments are below:

1. The MS should be revised by native english speaker.

2. line 58, add rodent-associated before viruses.

3. lines 74 and 77, The Yichun region? or The Yichun city?

4. line 106, add collection after Samples

5. line 143 and others, the primers sequences should be listed in a table.

6. Fig 2C and 2D, virus infection in rodents and humans detected in this study should be presented in another figure.

7. line 253, P should be itanlicized.

8. line 156, add S before segment

9. line 263, were the real-time RT-PCR products sequenced?

10. line 265, were Dabieshan orthohantavirus (DBSV) identified by primers for HTNV or SEOV? please clarify.

11. lines 276-277, "The cytochrome b gene sequences of rodents were deposited into GenBank under accession no. OQ187773-OQ187792." should be shown in line 240. In line 240, the rodent species idenfied by cytb gene should be adeed.

12. The description for genetic analysis and phylogenetic trees should be revised.

13. rodent name: full genus name at first mention, later with abbreviated genus name throughout the text (exception: at the beginning of a sentence)

14. Partial S should be amplified by PCR. Particularly, the genetic characteristic of SEOV infected in humans should be clarified.

We cannot make any decision about publication until we have seen the revised manuscript and your response to the reviewers' comments. Your revised manuscript is also likely to be sent to reviewers for further evaluation.

Sincerely,

Bruce A. Rosa

Academic Editor

Andrea Marzi

Section Editor

Both reviewers have requested substantial additional details for methodology, which would be necessary for a proper evaluation of the results after revision. Adding more detail in the discussion of the biology of the hosts and transmission in the introduction and discussion would also help to strengthen the paper. Reviewer 2 has provided more detailed comments directly in the manuscript file, so please ensure that you review those carefully. The authors do not need to strictly adhere to every suggestion about which sentences belong in which sections, but please consider the revisions outlined. Please also consider asking a native english speaker to revise the manuscript for grammar, as suggested by reviewers. 

Reviewer 1's additional comments are below:

1. The MS should be revised by native english speaker.

2. line 58, add rodent-associated before viruses.

3. lines 74 and 77, The Yichun region? or The Yichun city?

4. line 106, add collection after Samples

5. line 143 and others, the primers sequences should be listed in a table.

6. Fig 2C and 2D, virus infection in rodents and humans detected in this study should be presented in another figure.

7. line 253, P should be itanlicized.

8. line 156, add S before segment

9. line 263, were the real-time RT-PCR products sequenced?

10. line 265, were Dabieshan orthohantavirus (DBSV) identified by primers for HTNV or SEOV? please clarify.

11. lines 276-277, "The cytochrome b gene sequences of rodents were deposited into GenBank under accession no. OQ187773-OQ187792." should be shown in line 240. In line 240, the rodent species idenfied by cytb gene should be adeed.

12. The description for genetic analysis and phylogenetic trees should be revised.

13. rodent name: full genus name at first mention, later with abbreviated genus name throughout the text (exception: at the beginning of a sentence)

14. Partial S should be amplified by PCR. Particularly, the genetic characteristic of SEOV infected in humans should be clarified.

Reviewer's Responses to Questions

**Key Review Criteria Required for Acceptance?**

**Methods**

-Are the objectives of the study clearly articulated with a clear testable hypothesis stated?

-Is the study design appropriate to address the stated objectives?

-Is the population clearly described and appropriate for the hypothesis being tested?

-Is the sample size sufficient to ensure adequate power to address the hypothesis being tested?

-Were correct statistical analysis used to support conclusions?

-Are there concerns about ethical or regulatory requirements being met?

Reviewer #1: Partial S should be amplified by PCR. Particularly, the genetic characteristic of SEOV infected in humans should be clarified.

Reviewer #2: -The article lack a clear objective. 

-The methods are incomplete.

-Some methodology are write in the Introduction and in Results.

-The statistical analysis are not included in methods.

**Results**

-Does the analysis presented match the analysis plan?

-Are the results clearly and completely presented?

-Are the figures (Tables, Images) of sufficient quality for clarity?

Reviewer #1: (No Response)

Reviewer #2: -the epigraphs of tables and figures are not right writes.

-Some results are not presented in methods.

**Conclusions**

-Are the conclusions supported by the data presented?

-Are the limitations of analysis clearly described?

-Do the authors discuss how these data can be helpful to advance our understanding of the topic under study?

-Is public health relevance addressed?

Reviewer #1: (No Response)

Reviewer #2: The public health relevance is clear but the biology of the hosts, the importance of the different rodent species in the transmission, how are the transmission to human and other important topics are lacked in the discussion that help to understand the dynamic of the HFRS and how could prevent it.

The discussion are incompleted.

**Editorial and Data Presentation Modifications?**

Reviewer #1: (No Response)

Reviewer #2: (No Response)

**Summary and General Comments**

Reviewer #1: (No Response)

Reviewer #2: The topic of this manuscript is relevant to the public health, but the form to expose the analyses, data and results require a lot of changes because the sentences are write in not correct sections.

Additionally, there are not a clear objetive, the introduction is so short (lack of information detailed in the attach revised file) and Methods and Result have to improve.

The discussion have to improve too, adding more biological informations about the hosts.

All specific comments are added in the attach file.

I consider that the authors have to rewrite the manuscript for the readers can extract the valuable of their data (that are truly valuable).

PLOS authors have the option to publish the peer review history of their article (what does this mean?). If published, this will include your full peer review and any attached files.

Reviewer #1: No

Reviewer #2: No
---

## [Decision Letter · Decision Letter 1]

21 Jun 2023

Dear Ms Liu,

Thank you very much for submitting your manuscript "Orthohantavirus infections in humans and rodents in the Yichun region, China, from 2016 to 2021" for consideration at PLOS Neglected Tropical Diseases. As with all papers reviewed by the journal, your manuscript was reviewed by members of the editorial board and by several independent reviewers. The reviewers appreciated the attention to an important topic. Based on the reviews, we are likely to accept this manuscript for publication, providing that you modify the manuscript according to the review recommendations. 

The reviewers have had a chance to review the revised manuscript. They are mostly positive about it now, but there are just a few revisions suggested from reviewer 3 that should be considered, including trying to look at amino acid sequence changes, and more clearly stating the public relevance of the findings.

Sincerely,

Bruce A. Rosa

Academic Editor

Andrea Marzi

Section Editor

The reviewers have had a chance to review the revised manuscript. They are mostly positive about it now, but there are just a few revisions suggested from reviewer 3 that should be considered, including trying to look at amino acid sequence changes, and more clearly stating the public relevance of the findings.

Reviewer's Responses to Questions

**Key Review Criteria Required for Acceptance?**

**Methods**

-Are the objectives of the study clearly articulated with a clear testable hypothesis stated?

-Is the study design appropriate to address the stated objectives?

-Is the population clearly described and appropriate for the hypothesis being tested?

-Is the sample size sufficient to ensure adequate power to address the hypothesis being tested?

-Were correct statistical analysis used to support conclusions?

-Are there concerns about ethical or regulatory requirements being met?

Reviewer #1: (No Response)

Reviewer #3: The objectives of the study articulates the testible hypothesis. Study design is appropriate.

It remains unclear what was the reason for using neighbor-joining method for phylogenetic analysis?

Sample size is sufficient to address the hypothesis.

**Results**

-Does the analysis presented match the analysis plan?

-Are the results clearly and completely presented?

-Are the figures (Tables, Images) of sufficient quality for clarity?

Reviewer #1: (No Response)

Reviewer #3: 1. Analysis of amino acid sequences would be more informative compared to nucleotide sequences. That is because some nucleotide substitutions may not change amino acid resulting in limited changes in protein function. Also, without comparison of amino acid sequences the statement of finding new lineages of Hantaan orthohantavirus has limited support.

2. Strain SGHu02/2020 has different location on S and M segment phylogenetic tree. How authors could explain this.

**Conclusions**

-Are the conclusions supported by the data presented?

-Are the limitations of analysis clearly described?

-Do the authors discuss how these data can be helpful to advance our understanding of the topic under study?

-Is public health relevance addressed?

Reviewer #1: (No Response)

Reviewer #3: Public relevance should be clearly stated.

**Editorial and Data Presentation Modifications?**

Reviewer #1: (No Response)

Reviewer #3: (No Response)

**Summary and General Comments**

Reviewer #1: (No Response)

Reviewer #3: (No Response)

PLOS authors have the option to publish the peer review history of their article (what does this mean?). If published, this will include your full peer review and any attached files.

Reviewer #1: No

Reviewer #3: No

Figure Files:

Data Requirements:

Reproducibility:

References

---

## [Editor Report · Decision Letter 2]

18 Jul 2023

Dear Ms Liu,

We are pleased to inform you that your manuscript 'Orthohantavirus infections in humans and rodents in the Yichun region, China, from 2016 to 2021' has been provisionally accepted for publication in PLOS Neglected Tropical Diseases.

Best regards,

Bruce A. Rosa

Academic Editor

Andrea Marzi

Section Editor

The authors have sufficiently addressed reviewer concerns.

---

## [Editor Report · Acceptance letter]

2 Aug 2023

Dear Ms Liu,

We are delighted to inform you that your manuscript, "Orthohantavirus infections in humans and rodents in the Yichun region, China, from 2016 to 2021," has been formally accepted for publication in PLOS Neglected Tropical Diseases.

Best regards,

Shaden Kamhawi

co-Editor-in-Chief

Paul Brindley

co-Editor-in-Chief
